# Systematic Review and Meta-Analysis on Optimal Timing of Surgery for Acute Symptomatic Metastatic Spinal Cord Compression

**DOI:** 10.3390/medicina60040631

**Published:** 2024-04-13

**Authors:** Nicola Bresolin, Luca Sartori, Giacomo Drago, Giulia Pastorello, Paolo Gallinaro, Jacopo Del Verme, Roberto Zanata, Enrico Giordan

**Affiliations:** 1Department of Neuroscience, University of Padua, 35123 Padua, Italy; 2Neurosurgical Department, Aulss2 Marca Trevigiana, 31100 Treviso, Italy

**Keywords:** MSCC, timely surgery, metastases, spine, neoplasms

## Abstract

*Introduction***:** Symptomatic acute metastatic spinal epidural cord compression (MSCC) is an emergency that requires multimodal attention. However, there is no clear consensus on the appropriate timing for surgery. Therefore, to address this issue, we conducted a systematic review and meta-analysis of the literature to evaluate the outcomes of different surgery timings. *Methods***:** We searched multiple databases for studies involving adult patients suffering from symptomatic MSCC who underwent decompression with or without fixation. We analyzed the data by stratifying them based on timing as emergent (≤24 h vs. >24 h) and urgent (≤48 h vs. >48 h). The analysis also considered adverse postoperative medical and surgical events. The rates of improved outcomes and adverse events were pooled through a random-effects meta-analysis. *Results***:** We analyzed seven studies involving 538 patients and discovered that 83.0% (95% CI 59.0–98.2%) of those who underwent urgent decompression showed an improvement of ≥1 point in strength scores. Adverse events were reported in 21% (95% CI 1.8–51.4%) of cases. Patients who underwent emergent surgery had a 41.3% (95% CI 20.4–63.3%) improvement rate but a complication rate of 25.5% (95% CI 15.9–36.3%). Patients who underwent surgery after 48 h showed 36.8% (95% CI 12.2–65.4%) and 28.6% (95% CI 19.5–38.8%) complication rates, respectively. *Conclusion***:** Our study highlights that a 48 h window may be the safest and most beneficial for patients presenting with acute MSCC and a life expectancy of over three months.

## 1. Introduction

The spine is a common site for cancers, often involving the vertebral bodies [1,2]. The thoracic level is the most frequent location (60–70%), followed by the lumbar (20–30%) and cervical (10%) regions [3]. A severe complication of spinal involvement in cancer is symptomatic metastatic spinal epidural cord compression (MSCC), which affects 5–10% of patients [4,5]. Symptoms can manifest suddenly within 48 h in up to 28% of patients, necessitating prompt intervention [6]. Standard treatments for acute MSCC include corticosteroids, radiotherapy, and decompressive surgery. Studies indicate that combining decompressive surgery with radiotherapy leads to higher post-interventional improvement and ambulation rates, even in non-ambulatory patients [7].

To address the need for effective treatment of symptomatic MSCC, it is crucial to consider the timing of interventions and its impact on patient outcomes. There is still a debate surrounding the appropriate timing for decompression, and few guidelines have considered the timing’s effect on the decision-making process for treating MSCC [8]. 

In referral tertiary or secondary hospitals with a dedicated spine team, it is expected to receive patients with acute symptomatic MSCC in the emergency room at any time. Some of these patients may have a known history of cancer, while others may not. Moreover, symptomatic MSCC is possibly an indication of a change in a previously stable oncological condition and thus requires a new round of investigations. The spine surgeon and other physicians are responsible for providing timely care to such vulnerable patients while considering all medical, anesthesiological, and surgical problems associated with fragile oncological patients. In addition, predicting survival and functional outcomes is essential when selecting individual treatment for patients with MSCC, as well as the quality of the patient’s remaining life, which is becoming a cornerstone aspect in guiding the therapeutic decision-making process [9]. MSCC is particularly challenging during nights or weekends when access to multimodal (i.e., oncological and radiological) consultation and advanced imaging may be limited [10]. 

In certain medical cases, deciding when to perform surgery can be a difficult task. While it is widely known that neurological deficits have specific time windows for better reversal, spine surgeons may encounter other factors that can impact the outcome of the surgery differently than simpler occurrences such as acute epidural compression due to hematoma or herniations. Therefore, it is crucial for surgeons to make an accurate decision on when to perform the surgery. Should the surgeon focus on urgent decompression or postpone the intervention until the patient is adequately studied and prepared? The timing of surgery is of utmost importance to consider. 

The objective of the study was to systematically review and conduct a meta-analysis of the literature on acute and symptomatic metastatic spinal cord compression (MSCC) from the past decade. The study aimed to analyze and describe the various outcomes and complication rates of treatment for acute MSCC, focusing on the timing of surgery. To achieve this, the study proposed a systematic review and meta-analysis to analyze the outcomes of emergent (≤24 h) vs. non-emergent (>24 h) and urgent (≤48 h) vs. non-urgent (>48 h, late) surgeries and assess the safety and feasibility of decompression/palliative surgery for acute symptomatic MSCC.

## 2. Materials and Methods

### 2.1. Meta-Analysis Study Selection Process

The meta-analysis was reported following the Preferred Reporting Items for Systematic Reviews and Meta-Analyses (PRISMA, 2009) [11]. Original articles on acute and symptomatic MSCC were extracted from the literature. A literature search of the Web of Science (WOS, Clarivate Analysis, Thomson Reuters), PubMed, Embase, and Scopus databases was conducted. The databases were searched from 1 January 2013 to 10 November 2023, using the terms “acute”, “spine”, “metastasis”, “MSCC”, “timely”, and “emergent”, alone or in combination. After reviewing titles and abstracts, original research was included in the meta-analysis process. Letters, case reports, meta-analyses, reviews, editorials, in vitro or cadaveric studies, and corrections were excluded. Additional papers were added after reference analysis and after manually exploring each study considered relevant.

The question posed in this meta-analysis was the following:

“What are the rates of improved outcomes and adverse events rates for different timing of palliative surgery for acute symptomatic MSCC?”

Initially, only titles were examined, and non-relevant articles were excluded. Subsequently, abstracts were reviewed, and the full text of any study of interest was obtained. The full text of each selected study regarding lumbar pathologies was analyzed by one of the investigators (E.G.), and only original research papers with an adequate description of the procedure, outcomes, and follow-up were included.

To capture the highest quality articles published, the inclusion criteria were as follows:Adequate cohort assessment:
-Age ≥ 18 years;-Patients with solid neoplasms with a known or unknown primary tumor;-Studies with contrast-enhanced spine magnetic resonance (MRI) and spine computer tomography (CT);-Patients with neurological impairment at admission (i.e., Frankel grade A to D and ASIA grade A to D);-Patients with >3 months of survival;-Decompressive/palliative surgery with or without posterior screw fixation.Outcome assessment:
-≥1 point of improvement on Frankel score [6];-≥1 point of improvement ASIA scale [7].Adequate number of patients and timing:
-Studies with patients with MSCC symptoms from 0 to 72 h.-Studies reporting outcomes based on timing of surgery from the beginning of symptoms as ≤24 h, >24 h, ≤48 h, and >48 h.-Minimum > 25 consecutive patients.

Based on these inclusion and exclusion criteria, seven articles were identified (Figure 1).

We abstracted the following information from each study: study design, journal quartile, mean age of patients, sex distribution, adverse events, re-operation rates, metastasis recurrence rates, and 30-day mortality rates. We considered a minimum sample size of 25 patients for inclusion in the meta-analysis.

Emergent surgery was considered when performed within 24 h of symptom onset, urgent decompression was considered when performed within 48 h of symptom onset, and late surgery was considered when decompression was performed more than 48 h after symptoms.

We chose specific time cut-offs for our investigation, as we believe that the window of 0–48 h is ideal for reversing neurological deficits. Within this time frame, 0–24 h is expected to be the most effective, while some authors have also reported that the optimal time frames are between 24–48 h and 48–72 h [12]. However, based on our literature review, the most commonly used time cut-offs are those mentioned above (i.e., ≤24 h, >24 h, ≤48 h, and >48 h).

The proportion of patients who showed improvement in each study was measured based on an improvement of at least one point on the Frankel or ASIA score. Major adverse event rates were calculated considering postoperative medical and surgical adverse events. Abstracted medical adverse events included percentages of postoperative deep vein thrombosis, myocardial infarction, pulmonary embolisms, stroke, etc. Nerve injury and wound dehiscence/discitis percentages were abstracted separately. Revision surgeries were included when a patient needed an additional intervention to control metastasis progression or incomplete decompression.

### 2.2. Evaluation of Methodological Quality for the Meta-Analysis

For each study, the risk of bias was assessed. A visualization tool (Robvis) was used to visualize the risk of bias in our systematic review and meta-analysis. The Robvis tool comprises five domains for assessing different aspects of study biases.

To help define the different domains of Robvis, studies were analyzed based on the following criteria:Did the study include all or consecutive patients with adequate clinical follow-up?Was the outcome assessment objective and replicable?Was the sample size enough to draw valid statistical and clinical predictions?

Studies judged to have a low risk of bias were defined as those with a predefined study protocol (retrospective vs. prospective), adequate clinical follow-up (≥6 months vs. <6 months), objective and replicable assessment scales (i.e., Frankel, ASIA), and adequate cohort size (>25 vs. >100 patients). 

Risk-of-bias results are summarized in Table 1.

### 2.3. Statistical Analysis

Descriptive statistics were reported as median and interquartile range for continuous variables, and proportions and percentages for categorical variables. Parametric comparisons were conducted using Student’s *t*-test, while nonparametric comparisons were performed using the Mann–Whitney U test.

In the studies pooled in the meta-analysis process, the proportion of patients considered improved by at least 1 point in ASIA or Frankel score for each cohort, as well as the proportion of patients experiencing adverse events, was estimated. 

The rates of events were pooled across studies using the DerSimonian and Laird random-effects models [20]. Anticipating heterogeneity between studies, this model was chosen a priori because it incorporated both within- and between-study variances. Additionally, the Freeman–Tukey double arcsine transformation was utilized because the outcome rate was close to 0 or 1 in some studies.

All statistical tests were two-tailed, and the alpha (α) level was set at 0.05. Analyses were performed using commercially available software (Stata 13.0, StataCorp, College Station, TX, USA).

## 3. Results

Seven retrospective articles were identified, encompassing a total of 538 patients. Urgent surgical treatment (≤48 h) was delivered in 222 patients across five studies, while delayed treatment (>48 h) was provided to 197 patients across the same number of studies. Additionally, emergent treatment (≤24 h) was reported in 270 patients across four studies, while delayed treatment (>24 h) was provided to 212 patients across six studies. Two studies included the <24 h and <48 h population of patients [15,19]. Those populations were abstracted separately and pooled in the relative time frame subgroups. The minimum postoperative follow-up period ranged from 2 to 12 months across the included studies.

The characteristics of the included articles are summarized in Table 2.

The mean age of patients ranged from 56.8 to 66.1 years, with 39.9% of patients being female. The primary tumor distribution at presentation was as follows: prostate (26.4%), lung (21.7%), breast (12.6%), gastrointestinal tract (7.5%), renal (3.3%), and liver (2.1%), with other solid tumors accounting for the remaining 21.1%. All patients presented with neurological symptoms (i.e., Frankel or ASIA scores A to D), with 56.9% being severely impaired or non-ambulant (i.e., Frankel or ASIA scores A to B). The primary neoplasm origin was unknown in 5.3% of patients. Mean follow-up duration ranged from 2 to 6 months.

Only one study was considered at high risk of bias (Table 2).

### Outcomes and Adverse Events

All the studies provided reliable data regarding improved outcomes and adverse event rates. Overall, the post-operative improvement rate was 56.4% (95% CI 35.7–77.6%), the major adverse events rate was 19.8% (95% CI 5.1–40.3%), and the postoperative epidural hematoma rate was 1.2% (95% CI 0.0–3.5%), while the post-surgery infection rate was 6.5% (95% CI 1.2–15.0%) and the reoperation rate was 1.4% (95% CI 0.0–4.4%). Overall, the 30-day mortality rate was 0.1% (95% CI 0.0–0.9%). 

After stratifying the analysis by timing, it was found that the percentage of patients with an improved neurological symptom rate was 41.3% (95% CI 20.4–63.3%) when operated on within ≤24 h, while the complications rate was 25.5% (95% CI 15.9–36.3%), with epidural hematoma rates ranging of 3.6% (95% CI 0.6–17.7%) and wound dehiscence/discitis rates of 7.1% (95% CI 1.9–10.0%). In patients undergoing surgery after >24 h, the percentage of improved neurological symptoms was 32.2% (95% CI 12.8–79.9%), while the complication rate was 28.6% (95% CI 19.5–38.8%). 

After stratifying the analysis between surgery within 48 h and later than 48 h, it was found that the rate of patients with improved neurological symptoms was 83.0% (95% CI 59.0–98.2%) when operated within ≤48 h, although the adverse events rate was 21.0% (95% CI 1.8–51.4%) and the reoperation rate was 1.8% (95% CI 0.0–5.4%). The wound dehiscence/discitis rate was 7.9% (95% CI 0.5–20.8%), and the postoperative epidural hematoma rate was 1.7% (95% CI 0.0–5.5%). In patients undergoing surgery after > 48 h, the improved neurological symptoms rate was 36.8% (95% CI 12.2–65.4%), while the complications rate was 28.6% (95% CI 19.5–38.8%) and the wound dehiscence rate was 16.1% (95% CI 9.3–24.3%). Almost half (46.9%) of non-ambulant patients at presentation recovered to ambulation when operated within 48 h.

A significant number of patients showed improved outcomes after undergoing urgent decompression surgery, as compared to those who underwent emergent or late surgery (*p*-value < 0.001). Although the complication rates did not show any significant difference between urgent and emergent decompression (*p*-value: 0.241), they were lower in urgent decompression than late decompression (*p*-value: 0.030).

Data on outcomes and complications are summarized in Table 3.

## 4. Discussion

To the best of our knowledge, this study presents the most current and comprehensive review and meta-analysis on outcomes and adverse events associated with different treatment timings for symptomatic MSCC [13,14,15,16,17,18,19]. Our analysis indicates that urgent decompression, defined as within 48 h of symptom onset, results in significantly better neurological outcomes for patients with a life expectancy of over three months. Additionally, the complication rates associated with urgent decompression are comparable to those of emergent decompression and lower than those observed in patients undergoing surgery later.

Our findings indicate that more than 80% of patients who underwent urgent decompression reported some strength improvement. Furthermore, over 45% of nonambulatory patients regained the ability to walk upon discharge, representing a substantial improvement compared to approximately 40% of patients who experienced some degree of improvement after emergent decompression and 36% of those who underwent late decompression (>48 h). Additionally, overall complications were slightly higher in the >48 h group of patients compared to the ≤48 h group, in line with the rates already reported in the literature for elective MSCC surgeries [21].

According to our study, it is best to schedule surgery within 48 h of symptom presentation for MSCC. This is because patients can be better prepared for the operation while the neurological window for potential improvement is still open during this period [22]. This applies not only to the surgery itself but also to the need for an optimized radiological and medical preoperative workout, a multimodal pathway, and well-established communication lines between professionals, such as the emergency physician, medical oncologist, radio-oncologist, and spinal surgeon [22].

Occasionally, symptomatic MSCC can occur as the first symptom of cancer that has spread to other parts of the body [23]. This transpires in approximately 5% of the cases, as is also highlighted by our analysis [23]. Under such circumstances, it is crucial to consider the potential advantages and disadvantages of performing emergent surgery to reverse any neurological deficits in the setting of an unknown primary tumor site [24]. Indeed, it is imperative to ensure that the patient is well-prepared, beginning with a radiological diagnosis [10]. To minimize the risk of postoperative complications, it is essential to avoid surgery until the patient has undergone appropriate imaging (such as contrast-enhanced CT or MRI) and laboratory tests. These tests are necessary to rule out any underlying conditions or differential diagnoses caused by the primary tumor [25].

Although decompressive surgery, with or without additional spine stabilization, is generally considered a safe and feasible procedure when performed by experts, it may be associated with uncontrolled bleeding due to the nature of the tumor. Moreover, in cases where primary spine tumors are misdiagnosed, it could lead to inappropriate decompression, thus increasing the likelihood of tumor recurrences and decreasing overall neoplasm control [24].

Performing necessary examinations and gathering clinical information within 48 h may be adequate to make informed decisions. It is also advisable to consult with an oncologist to evaluate the risks and benefits of surgery and estimate the patient’s life expectancy. Therefore, the 0-to-24 h window following symptom onset may be less beneficial regarding risk reduction for the patient [13,15]. Our research revealed that patients treated in ≤48 h had a lower overall complication rate (21.0%) compared to those treated in ≤24 h (25.5%) or over 48 h (28.8%). This difference was particularly significant in terms of wound problems such as dehiscence or discitis, with a rate of 7.9% for patients treated ≤48 h compared to 16.1% for those treated >48 h. We speculate that this may be due to the failure to revert the neurological condition and the patient’s consequent inability to regain lower limb strength and walk, leading to an increased risk of wound-related issues. Patients who are bedridden and unable to walk have a higher likelihood of experiencing complications, which could result in longer hospital stays and a decreased quality of life. 

It is worth noting that the incidence of epidural hematoma is higher in patients who received surgery within 24 h (3.6%) compared to those who received surgery within 48 h (1.7%). This difference may be due to inadequate recognition of the metastasis’s nature and bleeding control, which can result from insufficient planning by the anesthesiologist and surgical team. Effective planning requires time and a multidisciplinary approach.

Several authors in the literature support our findings. Two studies highlight a slightly less favorable trend in patients undergoing emergent treatment than urgent treatment [13,15]. A Cochrane review, which included 544 patients, suggested that patients with acute symptomatic MSCC and an estimated survival of at least three months should be operated on within 48 h [26]. Furthermore, a randomized controlled trial found that patients with incomplete or progressive paraplegia for less than 48 h, who underwent decompressive surgery along with stabilization and radiotherapy, achieved a 62% post-treatment ambulatory rate [27]. In fact, some studies have shown that nonambulatory patients with deficits for less than 48 h before surgery consistently show improved mobility and ambulation [14,17]. Other authors have also reported that beneficial interventions for Frankel B patients and MSCC are most effective within 48 h after symptom onset. In particular, most patients showed neurological improvement of at least two grades on the Frankel scale and retained their sensory function [18].

## 5. Limitations

Our analysis has limitations as we were unable to collect individual patient data, which might have influenced our prognostic analysis with confounding and selection bias. That being said, only one study was found to be at high risk of bias. All the studies, despite being retrospective analyses, had adequate population characterization and outcome measures. However, in two of the studies, the 48 h cut-off included the time frame of less than 24 h, which may have led to selection bias. Separating the time frame of less than 24 h from the one of less than 48 h is a complex task, and no studies were found to make such a differentiation in the literature. Considering the strict selection criteria, it is possible to speculate that if the interventions were performed within 24 h, it would be specified in the studies, while if not, it would probably fall in the 24–48 h time window. Furthermore, outcomes were evaluated using a uniform grading system. This standardized grading system can be a valuable resource for conducting research to inform practice guidelines. Additionally, it may have implications for treatment and patient referral strategies within the hospital network, helping in the early identification and transfer of patients requiring prompt decompressive surgery.

## 6. Conclusions

According to this study, it is crucial for patients with localized spinal cord compression and a life expectancy of over three months to undergo urgent surgical decompression within 48 h of acute presentation without rushing for urgent intervention and allowing for adequate patient preparation in the first h after admission. This approach may result in significantly better neurological outcomes after surgery and can improve motor function, ambulation, and survival rate.

## Figures and Tables

**Figure 1 medicina-60-00631-f001:**
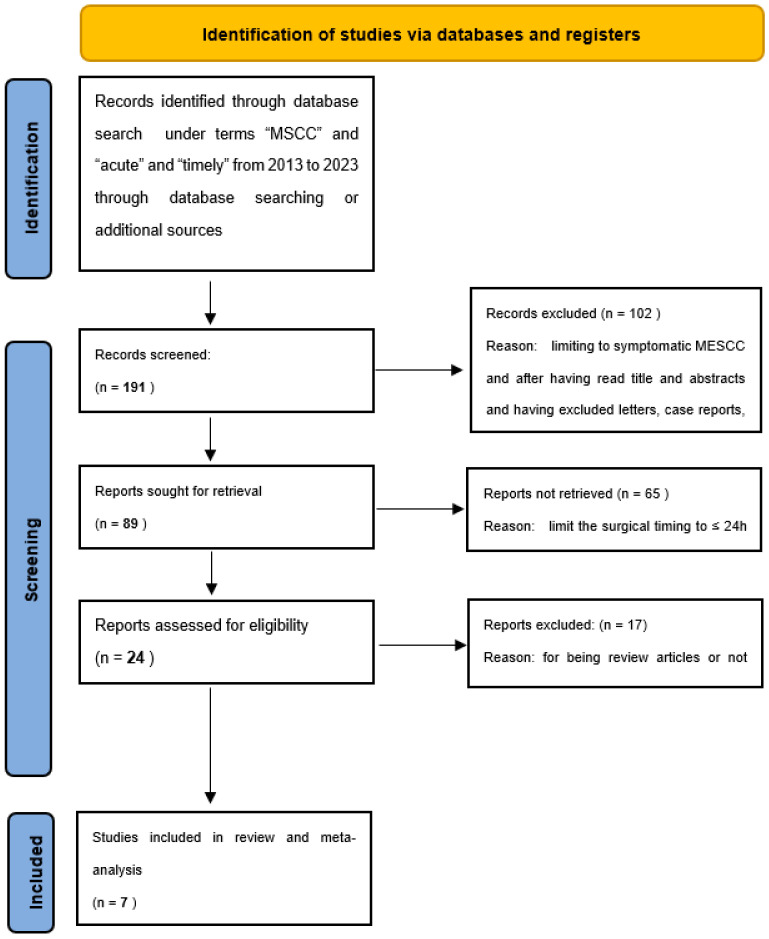
Flow diagram summarizing the process of study selection.

**Table 1 medicina-60-00631-t001:** Revised risk of bias tool for non-randomized trial.

Risk of Bias Domains
Study	D1	D2	D3	D4	D5	Overall
Meyer H et al., 2022 [13]						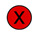
Younsi A et al., 2021 [14]	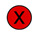			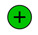	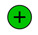	
Younsi A et al., 2020 [15]	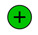	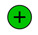			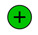	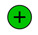
Tateiwa D et al., 2019 [16]		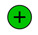			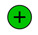	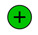
Lo WY et al., 2017 [17]		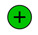		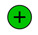		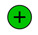
Fan Y et al., 2016 [18]		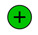		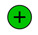	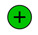	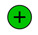
Quraishi NA et al., 2013 [19]	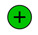	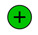				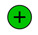

Domains: D1: Bias arising from the randomization process, D2: Bias due to deviations from intended intervention, D3: Bias due to missing outcome data, D4: Bias in measurement of the outcome, D5: Bias in selection of the reported result. Judgement: red circle—high; yellow circle—some concerns; green circle—low.

**Table 2 medicina-60-00631-t002:** Summary of the characteristics of the included studies.

Title	Author, Year	Journal	Recruitment	Country	Design	Cohort	Outcomes Scale	Numbers of Patients	Timing
Surgery in Acute Metastatic Spinal Cord Compression: Timing and Functional Outcome [13]	Meyer H et al., 2022	Cancers	2007–2019	Germany	Retro	Acute symptomatic MSCC, solid tumors, palliative surgery	ASIA score	46	≤24 h
Feasibility of salvage decompressive surgery for pending paralysis due to metastatic spinal cord compression [14]	Younsi A et al., 2021	Clinical Neurology and Neurosurgery	2004–2014	Germany	Retro	Acute symptomatic MSCC, solid tumors, palliative surgery	FRANKEL score	28	≤24 h
Impact of decompressive laminectomy on the functional outcome of patients with metastatic spinal cord compression and neurological impairment [15]	Younsi A et al., 2020	Clinical & Experimental Metastasis	2004–2014	Germany	Retro	Acute symptomatic MSCC, solid tumors, palliative surgery	FRANKEL score	101	≤48 h; >24 h
Clinical outcomes and significant factors in the survival rate after decompression surgery for patients who were non-ambulatory due to spinal metastases [16]	Tateiwa D et al., 2019	Journal of Orthopedic Science	2011–2016	Japan	Retro	Acute symptomatic MSCC, solid tumors, palliative surgery	FRANKEL score	31	≤48 h; >48 h
Metastatic spinal cord compression (MSCC) treated with palliative decompression: Surgical timing and survival rate [17]	Lo WY et al., 2017	Plos One	2012–2016	Taiwan	Retro	Acute symptomatic MSCC, solid tumors, palliative surgery	FRANKEL score	52	≤48 h; >48 h
The timing of surgical intervention in the treatment of complete motor paralysis in patients with spinal metastasis [18]	Fan Y et al., 2016	Eur Spine J	2007–2014	China	Retro	Acute symptomatic MSCC, solid tumors, palliative surgery	FRANKEL score	33	≤48 h; >48 h
Effect of timing of surgery on neurological outcome and survival in metastatic spinal cord compression [19]	Quraishi NA et al., 2013	Eur Spine J	2005–2010	United Kingdom	Retro	Acute symptomatic MSCC, solid tumors, palliative surgery	FRANKEL score	166	≤24 h ≤48 h; >48 h

**Table 3 medicina-60-00631-t003:** Summary of outcome and adverse events rates for lumbar FESS for different diseases.

	Overall	≤24 h	>24 h	≤48 h	>48 h
Patient N°	538	270	212	222	197
IMPROVED NEUROLOGICAL DEFICIT % (95% CI)	56.4 (35.0–77.6)	41.3 (20.4–63.6)	32.2 (12.4–55.8)	83.0 (59.0–98.2)	36.8 (12.2–65.4)
COMPLICATIONS % (95% CI)	19.8 (5.1–40.3)	25.5 (15.9–36.3)	28.6 (19.5–38.8)	21.0 (1.8–51.4)	28.6 (19.5–38.8)
WOUND DEHISCENCE/DISCITIS % (95% CI)	6.5 (1.2–15.0)	7.1 (1.9–10.0)	9.6 (1.8–21.2)	7.9 (0.5–20.8)	16.1 (9.3–24.3)
EPIDURAL HEMATOMA % (95% CI)	1.2 (0.0–3.5)	3.6 (0.6–17.7)	/	1.7 (0.0–5.5)	0.0 (0.0–3.6)
REOPERATION % (95% CI)	1.4 (0.0–4.4)	/	/	1.8 (0.0–5.4)	/
30 days MORTALITY % (95% CI)	0.1 (0.0–0.9)	/	0.1 (0.0–21.5)	0.1 (0.0–3.0)	/

Legend: CI: confidence interval, %: percentage, /: not enough data for meta-analysis.

## Data Availability

Data are contained within the article.

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
