# Peer review of "Systematic Review and Meta-Analysis on Optimal Timing of Surgery for Acute Symptomatic Metastatic Spinal Cord Compression"

_medicina, 2024, doi:10.3390/medicina60040631_

Round 1

Reviewer 1 Report

Comments and Suggestions for Authors

This paper is a systematic review of the timing of surgery for acute symptomatic spinal cord compression lesions

Although the article is original and has not been reported previously, it is not worthy of publication in that the following points are not clear.

The authors classify the timing of surgery for symptomatic spinal cord compression lesions into four groups: ≤ 24 hours, >24 hours, ≤ 48hours and >48hours.

In the first place, it is difficult to understand why the authors classified them in this way in their methodology.

What is the rationale for this classification?

Does the ≤ 48hours group not include patients classified as ≤ 24 hours?

Author Response

Reviewers 1 comments:
=====================
This paper is a systematic review of the timing of surgery for acute symptomatic spinal cord compression lesions

Although the article is original and has not been reported previously, it is not worthy of publication in that the following points are not clear.

The authors classify the timing of surgery for symptomatic spinal cord compression lesions into four groups: ≤ 24 hours, >24 hours, ≤ 48hours and >48hours.

In the first place, it is difficult to understand why the authors classified them in this way in their methodology. What is the rationale for this classification?

  • We thank the reviewers for their concerns. We apologize for the lack of clarity. A more detailed rationale was included in the introduction section (Page: 3; Paragraph: 3 Line: 1-8)

Does the ≤ 48hours group not include patients classified as ≤ 24 hours?

  • We appreciate reviewers' concerns and insight and apologize for the lack of clarity. We added a paragraph in the methods, results and limitation sections to better clarify such a point (Page: 7; Paragraph: 3, Line: 1-6 and Page 9, Paragraph 4, Line 5-7 and Page 14, Paragraph 3, Line 3-10 )

Reviewer 2 Report

Comments and Suggestions for Authors

This manuscript presents the results and conclusions that are based on a meta-analysis  which was centered on the determination of the optimal timing of surgery for acute symptomatic metastatic spinal cord compression. The collection of data is adequately described and well-presented, and the vast majority of bibliographic references are included in this review. The presentation of the results, as well as the relevant statistical analysis is adequate. 

This is a well organized retrospective meta-analysis which attempts to collect the relevant available data regarding the possible efficacy of the early (within 48 hours) decompression of the spinal cord from epidural compressive pathologies. Although most clinicians (orthopedics and neurosurgeons) adopt a similar clinical approach, this is a well supported review aimed to provide a theoritical support to the current surgical trend. Although ther are several inherent limitations in this manuscript, it could be concidered as scientifically sound.

Comments on the Quality of English Language

 The quality of English language is acceptable and only minor editing of English language is required.

Author Response

Reviewer 2
=====================

This is a well organized retrospective meta-analysis which attempts to collect the relevant available data regarding the possible efficacy of the early (within 48 hours) decompression of the spinal cord from epidural compressive pathologies. Although most clinicians (orthopedics and neurosurgeons) adopt a similar clinical approach, this is a well supported review aimed to provide a theoritical support to the current surgical trend. Although ther are several inherent limitations in this manuscript, it could be concidered as scientifically sound.

  • We thanks the reviewers for their insight and lovely comments.

Reviewer 3 Report

Comments and Suggestions for Authors

Congratulations for the paper. My comments are:

·       In the materials and methods section, the first paragraph should be deleted. The study objective should go at the end of the introduction section.

·       Table 1 should go in the results section.

·       Table 2 does not have text describing the table. Line 149.

·       The conclusion section should be written differently. The paragraph should comment: The conclusion of this study was.....

·       The style of references is not correct, according to Journal.

Author Response

=====================
Reviewier 3
=====================

Congratulations for the paper. My comments are:

  • In the materials and methods section, the first paragraph should be deleted. The study objective should go at the end of the introduction section.
  • We appreciate reviewers' suggestions and move it accordingly.

  • Table 1 should go in the results section.
  • We appreciate reviewers' suggestions, move it accordingly and reordered tables.

  • Table 2 does not have text describing the table. Line 149.
  • We appreciate reviewers' suggestions, moved and renamed it accordingly, and reordered tables.

  • The conclusion section should be written differently. The paragraph should comment: The conclusion of this study was.....
  • We appreciate the reviewers’s suggestions and reworded the conclusions (Page: 15; Paragraph: 2).

  • The style of references is not correct, according to Journal.
  • We apologize for the confusion and the inappropriate style of the journal. We changed the style to follow Journal guidelines.

Round 2

Reviewer 1 Report

Comments and Suggestions for Authors

I agree with the authors' opinion stated in the limitation, and I believe that the reasons for the inferior outcome in the group in which surgery was performed within 24 hours are still insufficiently discussed.

Author Response

Reviewer 1 comments:

I agree with the authors' opinion stated in the limitation, and I believe that the reasons for the inferior outcome in the group in which surgery was performed within 24 hours are still insufficiently discussed.

  • We thank the reviewers for their additional comments. We added an extra paragraph in the discussion for accounting for the inferior outcomes between groups (Page 14, Line 7-20)